# Comparison of Physical Activity and Sedentary Behaviour Patterns by Sex, Geographical Location, and Time of the Week in Mexican Adolescents

**DOI:** 10.3390/jfmk9040212

**Published:** 2024-10-30

**Authors:** Edtna Jáuregui-Ulloa, Julissa Ortiz-Brunel, Alejandro Gaytan-Gonzalez, Raúl Soria-Rodríguez, José Marcos Pérez-Maravilla, Martín Francisco González-Villalobos, Deborah Salvo, Darren E. R. Warburton, Juan Ricardo López-Taylor

**Affiliations:** 1Institute of Applied Sciences for Physical Activity and Sport (ICAAFyD), Department of Human Movement Sciences, Education, Sport, Recreation, and Dance, University Health Sciences Center (CUCS), University of Guadalajara, Guadalajara 44100, Mexico; edtna.jauregui@academicos.udg.mx (E.J.-U.); ortizbruneljulissa@gmail.com (J.O.-B.); raul.soria@academicos.udg.mx (R.S.-R.); jmarcos.perez@alumnos.udg.mx (J.M.P.-M.); martin.gvillalobos@academicos.udg.mx (M.F.G.-V.); 2Experimental Medicine Program, Faculty of Medicine, University of British Columbia, Vancouver, BC V6T 1Z4, Canada; alejandro.gaytang@ubc.ca (A.G.-G.); darren.warburton@ubc.ca (D.E.R.W.); 3Physical Activity Promotion and Chronic Disease Prevention Unit, Faculty of Education, University of British Columbia, Vancouver, BC V6T 1Z4, Canada; 4Department of Kinesiology and Health Education, The University of Texas at Austin, Austin, TX 78712, USA; dsalvo@austin.utexas.edu

**Keywords:** accelerometry, low–medium income country, movement behaviours, youths

## Abstract

**Background:** Excess sedentary behaviour (SB) and insufficient physical activity have been identified as risk factors for cardiometabolic diseases in adolescents, and some factors may affect how much time they spend on these activities. This study sought to compare the SB and PA patterns as well as compliance with PA recommendations by sex, geographical location, and time of the week in Mexican adolescents. **Methods:** In a cross-sectional design, we objectively assessed SB and PA in 106 adolescents (15 to 18 y) through waist-worn accelerometers for one week. The time spent in SB and in PA at different intensities was compared by sex, geographical location and time of the week with two-way and three-way repeated measures MANOVAs, while the compliance with physical activity recommendations (i.e., ≥60 min/day) was compared using chi-squared and McNemar tests. **Results**: Male participants spent more time in moderate, vigorous, and moderate to vigorous PA than females both during the whole week and on weekdays (all *p* < 0.05). There was no significant difference in SB nor PA by geographical location (i.e., metropolitan vs. non-metropolitan areas). Moreover, all participants spent more time on moderate, vigorous, and moderate to vigorous PA on weekdays than on weekends (*p* < 0.05). Compliance with international PA recommendations also showed a sex difference (males > females) and change between weekdays and weekends with no difference by geographical location. **Conclusions:** While geographical location does not seem to be a relevant factor, sex and time of the week appear to influence the SB and PA patterns in Mexican adolescents.

## 1. Introduction

Physical activity refers to any bodily movement that generates an energy expenditure according to intensity (level, moderate, and vigorous) [1]. Both aerobic (muscles are activating in a continued way, improving cardiorespiratory fitness) and muscle strengthening (to improve muscle fitness) physical activities provide physiological and psychological benefits; well, different intensities of these activities involve diverse positive health benefits [1]. On the other hand, a sedentary lifestyle is associated with chronic disease [1,2]. Moreover, sedentary behaviour (SB) has recently become more common in younger individuals [3,4], and adolescents do not usually meet the World Health Organization (WHO) physical activity guidelines [1,5]. In Mexico, this phenomenon was depicted in the most recent Mexican National Health and Nutrition Survey, where 42.5% of Mexican adolescents (15–19 years) did not meet the international WHO recommendations for physical activity (PA), and almost 91% of adolescents did not meet SB (screen time) guidelines [6,7].

The SB and PA patterns in adolescents can be influenced by several factors like sex, geographical location, and time of the week [8,9,10,11]. For instance, studies report differences by sex, where males tend to be more active than females [12,13]. Consistent evidence also indicates that adolescents in urban areas tend to be more active than those living in rural settings [8,14,15,16,17,18]. Nonetheless, the results regarding SB in urban vs. rural areas have been conflicting [14,15]. Regarding the time of the week, studies suggest that adolescents tend to be less active at the end of the week and weekends [14,15] compared to weekdays [14,19,20,21,22]. Similar results about these patterns have been reported in Mexico [6,23,24]. Although the results came from nationally representative samples, the information was collected from self-reported questionnaires [6,25] and the study that objectively measured SB and PA through accelerometers did not include a time of the week comparison [23]; thus, these patterns remain understudied in Mexican adolescents.

Since many of the behaviours acquired during adolescence tend to remain in adulthood [26,27,28,29,30], this period of life offers an opportunity for prevention and developing healthy lifestyle habits [30]. Thus, determining how much different factors affect the time adolescents spend in PA at different intensities and their SB can help identify areas for intervention in this population and develop public policies based on what factors may affect PA and SB the most in terms of the time spent in PA and SB and whether adolescents meet the international recommendations for PA.

Therefore, the purpose of this study was three-fold: (1) to compare the objectively measured time spent in SB and PA at different intensities by sex and geographic location; (2) to determine whether the SB and PA change between weekdays and weekends by sex and geographic location; and (3) to determine the proportion of compliance with the international recommendations of physical activity in high school students. We hypothesized that all three explanatory variables would affect SB and PA with higher scores in males, urban areas and weekdays in Mexican adolescents.

## 2. Materials and Methods

### 2.1. Study’s Design and Participants

This study used a cross-sectional design to identify patterns of physical activity and sedentary behaviours in adolescents at the high school level in the state of Jalisco, Mexico. We aimed to recruit 300 participants following a two-stage non-probabilistic sampling design. We first identified all the high schools that belonged to the University of Guadalajara, Mexico public education system (*n* = 55); these high schools were located across the state of Jalisco, Mexico. The principal of each school was contacted, provided with a brief explanation of the study’s objective, and invited to include their high school in the study, of which only 10 agreed to participate (18%). In the second stage, we collected and combined the lists of the students who were registered in their first high school year and were between 15 and 19 years old, from which we took a random sample (i.e., *n* = 300) including both sexes. The research team met with the selected students and their parents to explain the study’s objectives and methods further. After this initial visit, 101 students were not included (100 refused to participate, and 1 used a wheelchair), leading to 199 students who agreed to participate and whose parents provided signed informed consent (66% of the expected sample); we did not include students with a physical disability or temporal injury that may affect their usual physical activity or sedentary behaviours. On the second visit, the participants had anthropometric measurements and answered socio-demographic questionnaires. On a third visit, participants started to wear the accelerometer. On the fourth visit, the research team collected the accelerometers from the participants after one week of use. The time between each visit was no longer than one week, except between the third and fourth visit, where more than one week was allowed depending on the logistics and because the data were already stored in the accelerometer. At the end of the study, 93 participants did not comply with the validation data for the accelerometer and were excluded from the analysis, leading to a sample of 106 students (35% of the expected sample) (Figure 1). The data collection took place in 2011.

### 2.2. Geographical Location

The high schools’ geographical location was categorized as Metropolitan Area (MA) if it was in the state’s capital or in one of the metropolitan municipalities around the state’s capital, and Non-Metropolitan Area (NMA) if it was located anywhere else in the state.

### 2.3. Anthropometric Measurements

Body height, mass, five skinfolds (triceps, biceps, subscapular, iliac crest, and calf), and two circumferences (waist and hip) were taken following the methodology of the International Society for Advancement of Kinanthropometry [31]. All anthropometrists were standardized in the technique, and they had a technical measurement error of ≤10% for skinfolds and ≤2% for body mass, height, and circumferences. Body height was measured in a standing position to the nearest 0.1 cm using a portable stadiometer SECA (Model BM214, Hamburg, Germany). Body mass was measured (to the nearest 0.5 kg) with a digital scale (Tanita HD-313, Japan); both height and weight were taken with participants standing on bare feet and wearing their usual clothes but reducing extra weight from jackets, belts and objects in their pockets. Skinfolds were measured with a Harpenden calliper with a 0.1 mm precision, while circumferences were measured with a metallic tape (Lufkin) to the nearest 0.1 cm. The five skinfolds were summed up to get an overall indicator of fatness. The waist-to-hip ratio was calculated as waist circumference/hip circumference. The body mass index (BMI), its age-specific z-score, and categorization were calculated and determined by entering the participants’ information into the WHO AnthroPlus software v.1.0.4 [32], which uses the World Health Organization references for growth for children and adolescents aged 5 to 19 years. The software uses the calculated z-score to categorize the BMI into one of the following categories: Thinness, ≤−2 SD; Normal weight, >−2 SD but <1 SD; Overweight, ≥1 SD but <2 SD; and Obesity, ≥2 SD [33].

### 2.4. Physical Activity

The physical activity of adolescents was measured with the ActiGraph GT3X+^®^ and GT3X+s accelerometers (ActiGraph LLC, Pensacola, FL, USA). The device was placed on the anterior superior thorn iliac and fastened with an elastic belt. The accelerometers were initialized at 30 Hz and were programmed to collect data for seven days (from Monday to Monday). Participants wore it over the right hip at all times except during water activities and night sleeping. Data in counts were downloaded at 60 s epoch. Compliance criteria for wearing accelerometers were based on the International Children’s Accelerometry Database (ICAD), which considered a valid day if it recorded at least 500 min of wearing time (i.e., ≥8.3 h), while non-wear time was defined as a period of at least 90 consecutive minutes with zero counts [34]. Given that the ICAD showed an average of 5.3 valid days, we decided to include the data of participants on at least four weekdays and one weekend day for the analysis. If a participant had five weekdays and two weekend valid days, we randomly selected four and one, respectively. We kept the number of valid days constant to compare the minutes spent during physical activity. Once we had the dataset with the valid cases, we used Evenson’s cut-off points to calculate the time spent at different physical activity intensities and in sedentary behaviour based on the counts per minute (CPM) [35]. Sedentary behaviour (SB) was considered for 0 to 100 CPM, light-intensity physical activity (LPA) was considered for 101 to 2295 CPM, moderate-intensity physical activity (MPA) was considered for 2296 to 4011 CPM, vigorous-intensity physical activity (VPA) was considered for ≥4012 CPM, and moderate-to-vigorous intensity physical activity (MVPA) was considered for ≥2296 CPM. We chose these cut-off points to keep our analysis consistent with those used in the Cooper study using the ICAD [36].

To make weekdays and weekend days’ time comparable, we calculated the average time spent in SB and PA on weekdays (i.e., time spent/4), leading to the time spent on an average weekday. Results were illustrated by geographical location variables and sex.

Compliance with the international physical activity recommendations was analyzed in three ways. We first identified how many days the participants met the WHO physical activity recommendation of ≥60 min a day of MVPA [1]. Then, we calculated how many participants reached ≥ 300 min of MVPA during the assessed week (i.e., 60 × 5, the recommendation in a cumulative way) and how many participants met the ≥60 min a day of MVPA by the time of the week based on their weekend day and average weekday.

### 2.5. Statistical Analyses

The continuous demographic variables were reported as mean ± standard deviation, and their distribution was analyzed using the Shapiro–Wilk test and by examining skewness and kurtosis. The variables that showed a normal distribution were compared by sex and geographical location with independent samples *t*-test, while the no normally distributed variables were compared by using the Mann–Whitney U-test. The categorical variables were reported as frequencies and percentages and were compared by sex and geographical location using the chi-squared test of independence.

We compared the time spent in SB and PA at different intensities in minutes by sex and geographic location with a two-way MANOVA (2 * 2, sex by geographic location) for the whole week and three-way repeated measures MANOVA (2 * 2 * 2, sex by geographic location by time of the week) for weekdays and weekend days using the following models:

Two-way MANOVA
Ey = b0 + b1 * S + b2 * G + b3 * S *G

Three-way repeated measures MANOVA
Ey = b0 + b1 * S + b2 * G + b3 * S * G + b4 × T + b5 * S × T + b6 * G * T + b7 * S * G * T
where Ey is the estimated score from the MANOVA, b0 is the intercept, b1 are the coefficients associated with each variable, S is the sex, G is the geographical location, T is the time of the week, and the remaining are the interaction terms for these variables. After running the MANOVAs, we carried out their corresponding factorial ANOVAs with the same design and models as the MANOVAs to identify where the differences were, if any.

The MANOVA results were reported with Wilk’s lambda (λ) and F-score, the corresponding group comparisons were reported as least squares means ± standard error of the mean (SEM), whereas the weekends vs. weekdays differences were reported as mean difference and 95% CI. We included omega squared (ω^2^) as the effect size statistic for group comparisons. The effect size was considered small, medium, or large for ω^2^ as 0.01, 0.06, and 0.14, respectively [37]. Any result below the small effect was considered negligible.

The compliance with physical activity recommendations on the average weekday was compared to the compliance on the weekend day with the McNemar test for paired proportions.

The results were deemed statistically significant at an alpha level of 0.05. All analyses were conducted using the SPSS Software (version 29.0.2.0 for Windows, IBM Corp., Armonk, NY, USA), and graphs were drawn in GraphPad Prism (version 10.1.2 for Windows, GraphPad Software, Boston, MA, USA). We used Mendeley v1.19.8 reference management software for the administration of corrected references and Grammarly Pro 2024 for the corrected punctuation.

## 3. Results

### 3.1. Demographics

Table 1 presents the participants’ characteristics by sex and geographic location. There was no statistically significant difference in age, geographical location, or hip circumference between males and females. However, male participants were statistically significantly heavier (both by body weight and BMI), taller, leaner (sum of five skinfolds), and had a larger waist circumference and waist-to-hip ratio than female participants. Similarly, male participants showed a lower proportion of normal BMI and higher on overweight than female participants. None of the demographics showed a statistically significant difference when compared by geographical location.

### 3.2. Physical Activity by Sex

Total wearing time and time spent on SB and PA at different intensities in minutes by sex are shown in Table 2. The MANOVA suggested that there was a statistically significant effect of sex (Wilk’s λ = 0.80, F_(4, 99)_ = 6.2, *p* < 0.001) for the six variables in the whole week analysis. Non-statistically significant differences were observed regarding wearing time, SB, or LPA. However, male participants spent more time in MPA, VPA, and MVPA than females, with MPA showing a small to medium effect size, while VPA and MVPA showed a large effect size.

When weekdays and weekend days were compared, the MANOVA also suggested that there was a statistically significant effect of sex (Wilk’s λ = 0.76, F_(4, 99)_ = 7.7, *p* < 0.001). Wearing time, SB, and LPA did not show statistically significant differences by sex on either time of the week. MPA (small to medium effect), VPA (large effect), and MVPA (medium to large effect) were significantly higher in males than females during weekdays, while MPA did not show a significant sex difference during weekend days; still, VPA (large effect) and MVPA (medium to large effect) were higher in males than females in weekend days.

### 3.3. Physical Activity by Geographical Location

Table 3 shows the time spent in SB and PA at different intensities in minutes, while the MANOVA suggested that there was no statistically significant effect of geographic location for the time spent in SB and different PA intensities for the whole week either in minutes (Wilk’s λ = 0.93, F_(4, 99)_ = 1.9, *p* = 0.119). The same pattern was observed for the analysis by time of the week in minutes (Wilk’s λ = 0.94, F_(4, 99)_ = 1.5, *p* = 0.203). Most comparisons showed a negligible effect, and a few showed a small to medium effect.

### 3.4. Interaction Between Sex and Geographical Location

Table 3 shows the comparisons where a statistically significant interaction was observed. Although the MANOVA did not detect a significant sex by geographical location interaction for the six variables for the whole week (Wilk’s λ = 0.94, F_(4, 99)_ = 1.5, *p* = 0.208), the ANOVA analyses showed a significant interaction on MPA (*p* = 0.035) and MVPA (*p* = 0.041) where males spent more time in MPA and MVPA than females in MA but not NMA high schools. Similarly, males in MA schools spent more time in MPA and MVPA than males in NMA schools, while no significant difference was observed between MA and NMA in females.

No statistically significant interactions between sex and geographical location were observed for the minutes spent on weekdays vs. weekends in the MANOVA (Wilk’s λ = 0.96, F_(4, 99)_ = 1.1, *p* = 0.341) or the ANOVA analyses (lowest *p* = 0.056). However, we reported the interaction for MPA (*p* = 0.056) and MVPA (*p* = 0.059) because the *p*-values were close to the significance threshold. In this analysis, the minutes spent in MPA and MVPA on an average day showed the same pattern as the whole week.

### 3.5. Physical Activity by Time of the Week

The MANOVA suggested that there was a statistically significant change in SB and PA at different intensities between weekdays and weekend days when the variables were expressed in minutes (Wilk’s λ = 0.76, F_(4, 99)_ = 8.0, *p* < 0.001), with no significant interaction with sex (minutes: Wilk’s λ = 0.99, F_(4, 99)_ = 0.1, *p* = 0.968); geographical location (minutes: Wilk’s λ = 0.98, F_(4, 99)_ = 0.6, *p* = 0.692); nor interaction between the three components (minutes: Wilk’s λ = 0.98, F_(4, 99)_ = 0.5, *p* = 0.756); moreover, the ANOVA analyses showed no significant interaction (lowest *p* = 0.201). Since no significant interaction was observed, the changes were reported for the aggregated data, and no differentiation was made by sex or geographical location.

The total wearing time did not change from the average weekday vs. the weekend day (mean change [95% CI], −2 min [−28, 24]; *p* = 0.900). The participants showed a non-significant change in SB (−1 min [−28, 26]; *p* = 0.946) and LPA (−16 min [−36, 3]; *p* = 0.097), but spent more time in MPA (10 min [6, 15]; *p* < 0.001), VPA (5 min [2, 8]; *p* < 0.001), and MVPA (16 min [9, 22]; *p* < 0.001) in the average weekday than the weekend day (Figure 2).

### 3.6. Compliance with Physical Activity Recommendations

The data showed that 43 participants (40.6%) did not meet the 60 min/day recommendation on any day, 27 (25.5%) only in one day, 17 (16.0%) in two days, 8 (7.5%) in three days, 9 (8.5%) in four days, and only 2 (1.9%) met the recommendations on all the five assessed days. Due to the low frequency observed in compliance for two or more days, we combined these categories before carrying out a statistical analysis by sex and geographical location. Table 4 shows these combined categories: the cumulative weekly compliance and the compliance by weekdays and weekends. All sex comparisons showed a statistically significant difference. Females showed a higher proportion of meeting the recommendation on zero days and a lower proportion of meeting the recommendation on two or more days than males. Similarly, females showed lower weekly compliance, weekdays and weekend days compliance than males. No significant differences were found for geographical location.

The compliance with the physical activity recommendation was higher on the average weekday (24.5%) than on the weekend day (12.3%) in the whole sample (difference [95% CI], 12.2% [3.3, 21.2]; *p* = 0.009). When the analysis was stratified by sex, males showed significantly higher compliance on the average weekday (39.2%) than on the weekend day (19.6%) (difference: 19.6% [5.2, 34.0]; *p* = 0.012), and females showed no statistically significant change from weekdays (10.9%) to weekends (5.5%) (difference: 5.4% [−5.1, 16.0]; *p* = 0.317). For the geographical location analysis, MA high schools showed a higher compliance on weekdays (29.2%) than weekends (15.4%) (difference: 13.8% [2.7, 25.0]; *p* = 0.020), while NMA high schools did not change from weekdays (17.1%) to weekends (7.3%) (difference: 9.8% [−5.1, 24.6]; *p* = 0.206).

## 4. Discussion

Studying the variability in physical activity intensity among adolescents during weekdays and weekends in MA and NMA settings can provide insights into the individual and environmental factors connected to adolescents’ physical activity patterns. Our study’s findings suggest that factors like sex and day of the week are linked to adolescents’ physical activity patterns and sedentary behaviours in Mexican adolescents. For instance, males engaged in more MVPA than females, and there is an overall trend of doing more MVPA on weekdays than weekends. The time spent in MVPA on the weekends seems to be an area for intervention, as noted by our results and those reported by Corder et al. in the SPEEDY study, where MPA and VPA tend to decrease through the year on weekends but remained relatively stable on weekdays [38]. These conclusions hold significant implications for programs and practices, underscoring the necessity for targeted interventions to promote physical activity, especially among adolescent females and on weekends [30]. Similarly, our study expands on presenting objectively measured physical activity data in a region outside Canada, the USA, and Europe to deepen the understanding of physical activity and sedentary behaviour patterns in diverse populations [39], which in turn may help develop strategies and physical activity related public policy tailored to the Mexican population.

It was observed that the geographic location, whether urban (MA) or rural (NMA), was not related to the average level of physical activity or sedentary behaviour determined by accelerometry among high school students. Similar findings were reported by the Mexican National Health and Nutrition Survey (ENSANUT), conducted in Mexico in 2016, which did not identify a statistically significant difference between children (*n* = 1843) and adolescents (*n* = 1440) in the urban vs. rural areas [24]. On the other hand, the HELENA-MEX study shows that adolescents living in urban areas generally displayed higher sedentary behaviour and better fitness and fatness profiles than their rural peers [23]. The different results found in these studies and ours could be attributable to different definitions of rural and urban areas. For example, the HELENA-MEX study included schools that seem more likely to be representative of rural areas than those in this study (i.e., non-metropolitan areas). Also, these patterns of variation may be explained by the change of the environments across the years; people in the NMA context usually were preserved of facilities to move to their needs, but with the passing of the years, rural environments are becoming similar to urban spaces; thus, people in rural context are near to the inactive levels of urbanized citizens.

The specific geographic location (when considering the urban vs. rural context) of a school can impact an enrolled student’s average level of physical activity [8,40]; for example, a comparison of various public schools in the southwestern United States observed a greater mean physical activity level in adolescents of moderately urban areas (8.17 min/d) vs. large urban areas (3.78 min/d). However, Franco Arevalo et al. observed an increase in a sedentary lifestyle in Spanish students during the transition from primary education to high school, with no significant difference between rural and urban communities [16]. The relationship between these patterns by geographic location can be attributed to several aspects, such as (a) urban areas having better infrastructure, more accessible access to fitness and sports facilities and social conditions and multipurpose urban environments [41], (b) physical activity being very often linked to social activities (school and work) in adolescents, which can explain why they are more active on weekdays [42], and (c) the PA of adolescents may depend on the influence of the family and the social and neighbourhood influences [43].

In our study, we found that the decrease in MVPA from weekdays to weekends was similar by sex and geographical location (i.e., no interaction), which contrasted with the results by Kallio et al., who reported that the declines in MVPA and increases in sedentary time were greater in boys than girls during the weekend days [44]. These sex differences can be explained by Rosenfeld [13], who reported that males are generally more active than females in almost every age category. They reported that males are motivated to exercise to prevent chronic degenerative diseases and to be more competitive. In contrast, females commonly perform physical activity for emotional support and a sense of well-being. Furthermore, Brazo-Sayavera et al. attributed sex differences mainly to sociocultural factors, where boys are more encouraged to participate in higher-intensity activities [12].

The distribution of physical activity levels during the week may be essential to determine adolescents’ movement patterns. Most adolescents prefer to be active during the week rather than on weekends [42]. This study found that adolescents were more active on weekdays than weekends, which may be influenced by socioeconomic factors, as previous studies reported in adults in the same context [41]. Studies in Mexico show that people are more active and use more active transportation because of work activities [41,45,46,47]. Another factor that can help explain our results that was not explored in this study is that adolescents are more active on weekdays because they have more social interactions with peers and more social activities influenced by the day of the week. Studies found that on Fridays, adolescents become less active [48].

Another explanation is that adolescents who attend school and receive more academic duties may have less time to be active outside the school [49], for example, during weekends. This is aligned with our results, where adolescents of both genders were more inactive during weekends than on weekdays. Nevertheless, we did not explore the duties load variable.

In this study, we have some limitations. Firstly, the sample included likely does not represent most adolescent Mexican students, mainly because the sampling happened in one state and the selection was not probabilistic. Similarly, the final sample for the analysis was considerably smaller than expected at the beginning (i.e., 35% of the initial sampling). Consequently, the mean age was nearly 15 years old, and the sample size was not enough to carry on analysis by age groups. Therefore, we cannot generalize our results for all age ranges. We decided to analyze one age group of 15 to 18 years to attenuate the lack of sample power.

Secondly, the geographical location was categorized according to the definitions of the educational system (i.e., whether it was located in a metropolitan area or not), and we used this as a proxy for urban and rural areas. However, the non-metropolitan areas might include characteristics of urban settings, which might explain the lack of differences by geographical location. Further studies should explore this possibility. First, different terms used to refer to rural and urban settings can affect the comparison of adolescents’ levels of movement behaviour patterns [8,18,38]. For example, rural regions can be defined as areas with a population of less than 10,000 [14] and urban areas with more than 10,000 in population. Rural areas may be micropolitan and rural [8], and urban areas may be large, medium, and small [17]. Consequently, exploring the geo-statistics will be very helpful because the income in the communities studied could affect physical activity levels in Mexican adolescents beyond those presented definitions.

Thirdly, we focused the analysis on three explanatory variables (i.e., sex, geographical location, and time of the week), which leaves room for residual confounding and alternative explanations to the observed differences or lack of difference. Our study did not assess socioeconomic status. This may be considered a limitation because, as we mentioned before, the environment may change over the years, and rural and urban environments might become similar. Rural settings are usually characterized by a lack of facilities to move, which may be directly related to socioeconomic status and the possibility of moving in similar urban ways. Approaching socioeconomic status may be a good factor to explain non-significant differences in both geographical location groups. Thus, further research is warranted to analyze a larger sample size and evaluate other factors like socioeconomic and psychosocial variables.

Finally, we focused our analysis on the participants’ current physical activity and compliance with international recommendations; however, this approach could be misleading since people may still benefit from small increases in physical activity and yet not reach the 60 min/day threshold [50]. We used this approach to keep the results in line with the current metrics to diagnose the effectiveness of public interventions in physical activity [51]. However, the promotion of physical activity should aim to increase the time spent at different intensities and not only to comply with a rigid threshold. Similarly, the intervention in one aspect of physical activity (e.g., increasing MVPA) may have beneficial effects by affecting another (e.g., reducing sedentary behaviour); this override phenomenon has been reported in children and adolescents [52] and deserves further investigation in the Mexican population.

Previous studies highlight the effects of socioeconomic conditions in adulthood that originated from childhood due to a lack of education and material resources [53]. At the same time, neighbourhood resources, such as exercise facilities, sports fields, and parks, motivate teens to perform more MVPA outside school hours [9]. Finally, it should be considered that social support and self-motivation can also be decisive factors to consider for promoting and continuing physical activity from childhood [54,55] and can be more determinant than the location area where the adolescents live.

## 5. Conclusions

In this study, we found that geographical location was not a relevant factor in observing differences in the time spent on SB or PA at any intensity. This may stand out as the fact that the environment is a modifiable factor, which means that it changes with the passing of the years. Previously, rural environments may imply moving a considerable distance to arrive for services and local needs, and with the lack of facilities to transport and access to these services, people must walk. Nowadays, rural environments have evolved and acquired facilities that provoke people to be more sedentary, similar to urban people. These environmental and social changes may impact and generate variations in physical activity and sedentary behaviour patterns that are different from some years ago; at least, this conclusion may apply to our study, providing substantial value to the existing body of knowledge. On the other hand, sex and time of the week seem more relevant in observing differences in PA of moderate, vigorous, and moderate to vigorous intensities, but not SB nor light PA intensity. Moreover, sex and time of the week consistently affected compliance with PA recommendations in Mexican adolescents. These results suggest that further interventions should aim to promote physical activity in adolescents, given the low compliance with international recommendations with a focus on female teenagers, where compliance is even lower. Finally, the time of the week should also be considered, as less compliance is observed during weekends. That is why we suggest encouraging physical activities designed for specific gender groups and weekend physical activities to increase the possibility of activating a similar population as our context.

## Figures and Tables

**Figure 1 jfmk-09-00212-f001:**
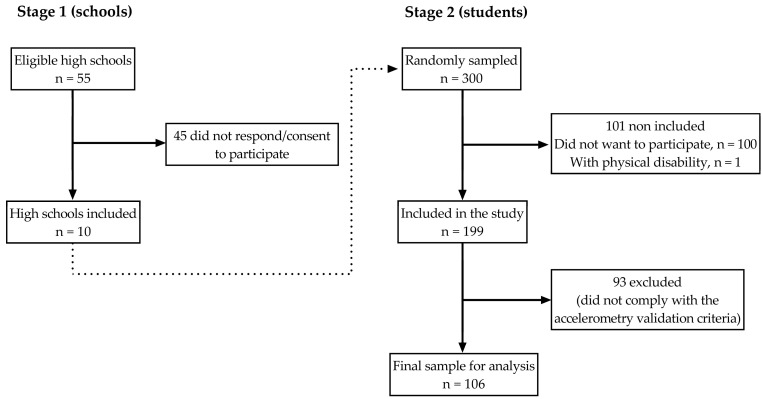
Flowchart depicting the selection process for the schools and participants. The dotted line indicates that the sample of 300 students was taken from the 10 high schools that agreed to participate.

**Figure 2 jfmk-09-00212-f002:**
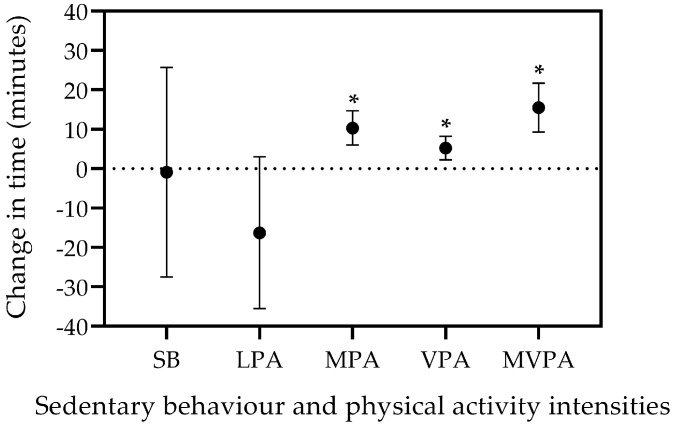
Changes in time spent in sedentary behaviour and physical activity at different intensities in minutes for the aggregated data. The differences were calculated as weekday–weekend days (i.e., a positive score indicates more time spent during weekdays). The circles represent the mean change, while the whiskers represent the 95% confidence interval for that change. * Denotes a statistically significant change in weekday vs. weekend days (*p* < 0.001). LPA: Light-intensity physical activity; MPA: Moderate-intensity physical activity: MVPA: Moderate-to-vigorous intensity physical activity: SB: Sedentary behaviour: VPA: Vigorous-intensity physical activity.

**Table 1 jfmk-09-00212-t001:** Participants’ characteristics divided by sex and geographical location.

	**Sex**		***p*-Value ^e^**
	**Males (*n* = 51)**	**Females (*n* = 55)**	
Age, y ^a^	15.7 ± 0.8	15.6 ± 0.7	0.403 ^f^
Geographical location ^b^			
MA	33 (64.7)	32 (58.2)	0.552 ^g^
NMA	18 (35.3)	23 (41.8)	
Height, cm	172.9 ± 5.8	160.2 ± 6.2	<0.001
Weight, kg	70.6 ± 14.0	56.7 ± 12.9	<0.001 ^f^
BMI, kg/cm^2^	23.5 ± 4.3	22.0 ± 4.5	0.031 ^f^
BMI categories			
Thinness	3 (5.9)	1 (1.8)	0.019 ^g^
Normal	25 (49.0)	43 (78.2) *	
Overweight	12 (23.5)	5 (9.1) *	
Obese	11 (21.6)	6 (10.9)	
Sum of five skinfolds, mm ^c^	71.6 ± 31.8	91.1 ± 32.3	0.002 ^f^
Waist circumference, cm	79.5 ± 9.6	70.6 ± 9.4	<0.001
Hip circumference, cm	96.2 ± 8.0	94.3 ± 9.1	0.093 ^f^
Waist-to-hip ratio	0.82 ± 0.05	0.75 ± 0.04	<0.001
	**Geographical Location**		***p*-Value**
	**MA (*n* = 65)**	**NMA (*n* = 41)**	
Age, y	15.3 ± 0.7	15.6 ± 0.7	0.827 ^f^
Sex			
Male	33 (50.8)	18 (43.9)	0.552 ^g^
Female	32 (49.2)	23 (56.1)	
Height, cm	166.6 ± 8.4	165.9 ± 9.3	0.719
Weight, kg	63.6 ± 14.2	63.1 ± 16.6	0.878
BMI, kg/cm^2^	22.8 ± 4.0	22.8 ± 5.1	0.982
BMI categories			
Thinness	1 (1.5)	3 (7.3)	0.327 ^g^
Normal	43 (66.2)	25 (61.0)	
Overweight	12 (18.5)	5 (12.2)	
Obese	9 (13.8)	8 (19.5)	
Sum of five skinfolds, mm ^d^	80.0 ± 29.8	84.2 ± 38.4	0.538
Waist circumference, cm	75.3 ± 10.0	74.3 ± 11.3	0.648
Hip circumference, cm	95.2 ± 8.2	95.2 ± 9.3	0.977
Waist-to-hip ratio	0.79 ± 0.06	0.78 ± 0.6	0.367

^a^ Reported as mean ± standard deviation. ^b^ Reported as frequency count (percentage). ^c^ Males *n* = 49, Females *n* = 52. ^d^ MA *n* = 61, NMA *n* = 40. ^e^ Calculated from independent samples *t*-test unless otherwise stated. ^f^ Calculated from Mann–Whitney U-test. ^g^ Calculated from a chi-squared test of independence. * Denotes statistically significant differences by sex within the specified category (*p* < 0.05). BMI: Body mass index; MA: Metropolitan area; NMA: Non-metropolitan area.

**Table 2 jfmk-09-00212-t002:** Wearing time, time spent in sedentary behaviour and in different intensities of physical activity for the whole week, weekdays, and weekends compared by sex (*n* = 106).

	Males (*n* = 51)	Females (*n* = 55)	Difference ^d^	*p*-Value	Effect Size ^g^
Whole week					
Wearing time, min ^a^	4610 ± 53	4616 ± 50	−6 ± 73	0.933 ^e^	0.000
SB, min	2745 ± 58	2884 ± 54	−140 ± 79	0.080	0.002
LPA, min	1600 ± 43	1554 ± 40	46 ± 58	0.433	0.000
MPA, min	174 ± 9	145 ± 9	28 ± 13	0.027	0.037
VPA, min	91 ± 9	32 ± 8	59 ± 12	<0.001	0.181
MVPA, min	265 ± 14	177 ± 13	88 ± 20	<0.001	0.153
Weekdays ^b^					
Wearing time, min	922 ± 11	922 ± 10	0 ± 15	0.988 ^f^	0.000
SB, min	548 ± 12	577 ± 11	−29 ± 16	0.070	0.022
LPA, min	318 ± 8	307 ± 8	11 ± 11	0.322	0.000
MPA, min	37 ± 2	31 ± 2	6 ± 3	0.050	0.027
VPA, min	19 ± 2	7 ± 2	12 ± 3	<0.001	0.146
MVPA, min	56 ± 3	38 ± 3	18 ± 4	<0.001	0.123
Weekend days ^c^					
Wearing time, min	921 ± 20	926 ± 18	−5 ± 27	0.848 ^f^	0.000
SB, min	553 ± 21	575 ± 19	−22 ± 28	0.432	0.000
LPA, min	328 ± 16	328 ± 15	0 ± 22	0.979	0.000
MPA, min	27 ± 3	21 ± 3	6 ± 4	0.146	0.011
VPA, min	14 ± 2	3 ± 2	11 ± 2	<0.001	0.165
MVPA, min	40 ± 4	24 ± 4	17 ± 5	0.002	0.078

^a^ Reported as least squares mean ± standard error of the mean; ^b^ time spent in one average weekday; ^c^ time spent in one weekend day; ^d^ some discrepancies are expected because of rounding. ^e^ Calculated from a two-way ANOVA for the effect of sex adjusted for geographical location for the whole week variables; ^f^ calculated from a three-way ANOVA for the effect of sex adjusted for geographical location and moment of the week for the weekday and weekend day variables; ^g^ calculated as omega squared. LPA: Low-intensity physical activity; MPA: Moderate-intensity physical activity; MVPA: Moderate-to-vigorous intensity physical activity; SB: Sedentary behaviour; VPA: Vigorous-intensity physical activity.

**Table 3 jfmk-09-00212-t003:** Comparisons that showed a significant interaction between sex and geographical location.

Variable	Subgroups ^a^	Comparison Within Subgroups ^b^	Mean 1 ^e^	Mean 2	Difference ^g^	*p*-Value
Whole week ^c^						
MPA, min	MA	Males vs. Females	199 ± 11 ^f^	144 ± 11	55 ± 16	<0.001
	NMA	Males vs. Females	149 ± 15	147 ± 13	1 ± 20	0.947
MVPA, min	MA	Males vs. Females	303 ± 17	175 ± 17	128 ± 24	<0.001
	NMA	Males vs. Females	226 ± 23	179 ± 20	47 ± 31	0.133
MPA, min	Males	MA vs. NMA	199 ± 11	149 ± 15	50 ± 18	0.008
	Females	MA vs. NMA	144 ± 11	147 ± 13	−4 ± 17	0.824
MVPA, min	Males	MA vs. NMA	303 ± 17	226 ± 23	77 ± 29	0.009
	Females	MA vs. NMA	175 ± 17	179 ± 20	−5 ± 27	0.864
Time of the week ^d^						
MPA, min	MA	Males vs. Females	36 ± 2	25 ± 2	11 ± 3	0.001
	NMA	Males vs. Females	27 ± 3	27 ± 3	1 ± 4	0.878
MVPA, min	MA	Males vs. Females	55 ± 3	30 ± 3	24 ± 5	<0.001
	NMA	Males vs. Females	42 ± 4	32 ± 4	10 ± 6	0.096
MPA, min	Males	MA vs. NMA	36 ± 2	27 ± 3	9 ± 4	0.025
	Females	MA vs. NMA	25 ± 2	27 ± 3	−1 ± 4	0.688
MVPA, min	Males	MA vs. NMA	55 ± 3	42 ± 4	13 ± 6	0.023
	Females	MA vs. NMA	30 ± 3	32 ± 4	−2 ± 5	0.751

^a^ Refers to the categories of Sex or Geographical location. ^b^ Refers to the categories being compared within each subgroup of Sex or Geographical location. ^c^ Corresponds to the aggregated time from the five assessed days (i.e., minutes per week). ^d^ Corresponds to the average assessed day from the average weekday and weekend day (i.e., minutes per day). ^e^ The mean 1 and mean 2 belong to the first and second categories, which are compared in the third column. ^f^ Data reported as least squares means ± SEM. ^g^ Some discrepancies are expected due to rounding. MA: Metropolitan area; MPA: Moderate-intensity physical activity; MVPA: Moderate-to-vigorous intensity physical activity; NMA: Non-metropolitan area.

**Table 4 jfmk-09-00212-t004:** Compliance with the physical activity recommendations compared by sex and geographical location.

	All	Sex			Geographical Location		
		Males	Females	*p*-Value ^e^	MA	NMA	*p*-Value ^e^
n	106	51	55		65	41	
Days of compliance ^a^							
Zero	43 (40.6)	13 (25.5)	30 (54.5) *		25 (38.5)	18 (43.9)	0.458
One	27 (25.5)	10 (19.6)	17 (30.9)	<0.001	15 (23.1)	12 (29.3)	
Two or more	36 (34.0)	28 (54.9)	8 (14.5) *		25 (38.5)	11 (26.8)	
Weekly compliance ^b^							
<300 min	85 (80.2)	33 (64.7)	52 (94.5)	<0.001	49 (75.4)	36 (87.8)	0.140
≥300 min	21 (19.8)	18 (35.3)	3 (5.5)		16 (24.6)	5 (12.2)	
Weekdays compliance ^c^							
<60 min	80 (75.5)	31 (60.8)	49 (89.1)	0.001	46 (70.8)	34 (82.9)	0.173
≥60 min	26 (24.5)	20 (39.2)	6 (10.9)		19 (29.2)	7 (17.1)	
Weekends compliance ^d^							
<60 min	93 (87.7)	41 (80.4)	52 (94.5)	0.037	55 (84.6)	38 (92.7)	0.362
≥60 min	13 (12.3)	10 (19.6)	3 (5.5)		10 (15.4)	3 (7.3)	

^a^ Represents the number of days the participant met the physical activity recommendations of at least 60 min of moderate-to-vigorous physical activity per day. ^b^ Represents whether the participant met the recommendation by accumulating the minutes for the whole week (i.e., 5 days * 60 min a day). ^c^ Represents whether the participant met the recommendation on an average weekday. ^d^ Represents whether the participant met the recommendation on the weekend day. ^e^ Calculated from a chi-squared test of independence for the comparison by sex and geographical location. * Denotes statistically significant differences by sex within the specified category (*p* < 0.05). MA: Metropolitan area; NMA: Non-metropolitan area.

## Data Availability

The data used for this article are available from the corresponding author upon reasonable request.

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
