# Peer review of "Comparison of Physical Activity and Sedentary Behaviour Patterns by Sex, Geographical Location, and Time of the Week in Mexican Adolescents"

_jfmk, 2024, doi:10.3390/jfmk9040212_

Round 1
Reviewer 1 Report
Comments and Suggestions for Authors
The manuscript is very interesting and comprehensive.
How was physical activity in water (e.g. swimming) calculated? Was there a diary in which they were described for activities not calculated by the device?
Author Response
Comment 1: “How was physical activity in water (e.g. swimming) calculated? Was there a diary in which they were described for activities not calculated by the device?”
Response 1: Thank you for your question. Our participants didn’t perform water activities during the measurement week, therefore any physical activities in water were not considered in the total calculation. We didn’t make any changes in the manuscript regarding this comment.
Reviewer 2 Report
Comments and Suggestions for Authors
I am writing to provide feedback on the recently reviewed scientific article titled " Comparison of physical activity and sedentary behaviour patterns by sex, geographical location, and time of the week in Mexican adolescents” (Manuscript Number: jfmk-3253380) submitted for publication Journal Functional Morphology and Kinesiology.
The article is well-written and provides a thorough description of the subject matter. However, it falls short in delivering innovative insights or conclusions that go beyond the obvious within the field of study.
Having 10 authors might be excessive. While the article is formally correct and meets all the journal's criteria, including an adequate, current, and well-founded literature review, there are some concerns. The study's database includes 106 adolescents, a relatively small sample size, which may pose challenges in generalizing the conclusions.
- A primary issue is that tables 2 and 3 present the same information (one in absolute values and the other in percentages), as do tables 4 and 5. Therefore, one of each pair should be removed.
- Additionally, in my opinion, none of the four tables provide substantial information to the article. Eliminating them might speed up the reading and improve the article's clarity. It may be necessary to expand the text slightly on page 5, lines 234-237, and page 6, lines 246-254.
Furthermore, the conclusions drawn in this article are unremarkable. From the outset, it was evident that the study's findings would lead to widely accepted conclusions within the discipline. This represents a missed opportunity to challenge prevailing assumptions, offer a fresh perspective, or engage in critical analysis. As a result, the article fails to add substantial value to the existing body of knowledge.
The study's conclusions are overly concise. Could you elaborate further and provide additional conclusions? Lastly, I noticed the absence of a discussion regarding the study's limitations. Could you please include this?
Author Response
Comment 1: “The article is well-written and provides a thorough description of the subject matter. However, it falls short in delivering innovative insights or conclusions that go beyond the obvious within the field of study”
Response 1: Thank you for pointing this out. We agree with your comment and we add the most important innovative insight into the conclusion section (page 16, lines 468-478) about how environments are changing and this may change the physical activity patterns to more sedentary behaviors, resulting in more similar rural locations to urban locations.
Comment 2: “Having 10 authors might be excessive”
Response 2: Thank you for your concern about authorship. All the authors contribute significantly to the manuscript. We are a multidisciplinary and interinstitutional research network that collaborates together since several years ago. We will keep the same authorship as the previous version.
Comment 3: “While the article is formally correct and meets all the journal's criteria, including an adequate, current, and well-founded literature review, there are some concerns. The study's database includes 106 adolescents, a relatively small sample size, which may pose challenges in generalizing the conclusions”
Response 3: Thank you for pointing this out, we agree with your comment. Therefore, we pointed out that our conclusions may explain and apply to our results, context, and sample in the conclusion section (page 19, lines 474-478).
Comment 4 “A primary issue is that tables 2 and 3 present the same information (one in absolute values and the other in percentages), as do tables 4 and 5. Therefore, one of each pair should be removed”
Response 4: Thank you for pointing this out. We agree with the comment. Therefore, we have deleted Table 3 and Table 5 from the manuscript; we leave a comment marked in red on page 9 in lines 251-252 and page 10 in lines 256-258 highlighting this. We change the number of table 4 to table 3 (page 9, line 253), table 6 to table 4 (page 10, line 260, page 11, line 303), and table 7 to table 5 (page 13, line 338, page 12, line 320) to have a logical sequence number of tables. Furthermore, we deleted from the methods the description about how we calculate the percentage of wearing time spent in sedentary behaviour, and in different intensities of physical activity for the whole week, weekdays, and weekends leaving relevant information from Table 2 and 4 (page 5 in lines 167-170, 186-190). Finally, we deleted lines in results paragraphs, figures and tables about the removed tables, final paragraphs without percentages results were found in page 6 lines 230-244, 246-247, page 10, lines 260-273, 275-290, page 12, lines 307-314. Finally, we delete information about percentages from the discussion section (page 17, line 417).
Comment 5: “Additionally, in my opinion, none of the four tables provide substantial information to the article. Eliminating them might speed up the reading and improve the article's clarity”
Response 5: Thank you for your comment, however we disagree. We consider that is important to show the results in that way and provide an easy way to consult the results of our study. We deleted the tables about percentages that were extra and unnecessary information as you suggested and we described the changes in the above comment.
Comment 6: “It may be necessary to expand the text slightly on page 5, lines 234-237, and page 6, lines 246-254"
Response 6: Thank you for your comment. We deleted lines 234-275 on page 5 because it was related to percentage results from Table 3 as a recommendation in another comment above; that’s why we didn’t expand the text. The same is the case for some of the lines 246-254, including results about percentages. We consider that sentences within these lines related to average minutes are well described and the lines about percentages were the lines that needed to expand the text, however, we deleted those results too. Therefore, we didn't consider it necessary to expand the text in these paragraphs
Comment 7: “Furthermore, the conclusions drawn in this article are unremarkable. From the outset, it was evident that the study's findings would lead to widely accepted conclusions within the discipline. This represents a missed opportunity to challenge prevailing assumptions, offer a fresh perspective, or engage in critical analysis. As a result, the article fails to add substantial value to the existing body of knowledge”
Response 7: Thank you for pointing this out, we agree. We add a more widely conclusion in this section (page 16, 469-479) that involves the substantial value to the existing body of knowledge denoting that environments are changing and this may cause pattern variations into sedentary tracks.
Comment 8: “The study's conclusions are overly concise. Could you elaborate further and provide additional conclusions?”
Response 8: Thank you for pointing this out, we agree. Therefore, we add more small conclusions about the change in the environment and how those changes are changing in turn physical activity patterns in conclusion sections (page 16, 469-479)
Comment 9: “Lastly, I noticed the absence of a discussion regarding the study's limitations. Could you please include this?”
Response 9: Thank you for pointing this out, we agree. We added some extra lines in discussion part (page 15, lines 440-446) related to the discussion of some limitations like the absence of socioeconomic status evaluation in our sample.
Reviewer 3 Report
Comments and Suggestions for Authors
This manuscript investigates the level of physical activity and sedentary behavior among Mexican adolescents, and establishes a comparison by age, geographic location (metropolitan/non-metropolitan), and day of the week (weekday/weekend). It finds that the level of physical activity (especially vigorous activity) is higher among boys, on weekdays, and shows no geographical differences.
The introduction begins with a brief description of how physical activity and sedentary behavior relate to health, and how patterns established during adolescence persist into adulthood. It would be advisable to delve deeper into this aspect, further differentiating by type and intensity of physical activity, and to also contextualize the research by referencing results from previous studies conducted in the region, such as those cited in the discussion.
The study is well designed, and the methodology is well described. However, for all the reasons mentioned in the methodology, the final sample size is small. It can be observed that the groups (by sex and geographic distribution) are adequately balanced, which lends credibility to the comparisons. Nonetheless, the sample size is objectively small, and it is also worth noting that although the target population is adolescents aged 15 to 18, the average age is very close to 15 years, which raises doubts about the representativeness of the sample and the possibility of drawing conclusions that can be generalized to a broader population.
Despite all of this, the results are presented clearly and in an organized manner, and some intervention proposals are derived. Here, again, it would be advisable to review the results obtained by other studies that have attempted to promote physical activity among adolescents, and perhaps adopt a more flexible perspective, considering other possible scenarios (Is it more necessary to try to encourage physical activity on weekends, or would it be more effective to promote it where it is already working, which is during the week?).
Author Response
Comment 1: “The introduction begins with a brief description of how physical activity and sedentary behavior relate to health, and how patterns established during adolescence persist into adulthood. It would be advisable to delve deeper into this aspect, further differentiating by type and intensity of physical activity, and to also contextualize the research by referencing results from previous studies conducted in the region, such as those cited in the discussion”
Response 1: Thank you for pointing this out, we agree. Therefore, we add physical activity definitions and concepts about the types and intensities of these activities on page 2 lines 42-54. Contextual previous studies in Mexico had been cited in the introduction, which we highlighted in this section (page 2, lines 63-68).
Comment 2: “The study is well designed, and the methodology is well described. However, for all the reasons mentioned in the methodology, the final sample size is small. It can be observed that the groups (by sex and geographic distribution) are adequately balanced, which lends credibility to the comparisons. Nonetheless, the sample size is objectively small, and it is also worth noting that although the target population is adolescents aged 15 to 18, the average age is very close to 15 years, which raises doubts about the representativeness of the sample and the possibility of drawing conclusions that can be generalized to a broader population”
Response 2: Thank you for pointing this out, we agree. Therefore, we add some lines about this limitation in the discussion section (Page 15, lines 421-424) highlighting that we decided to analyze a unique age group, because of the lack of the power that englobe the range group and the possibility of no analyze by age group getting representative results.
Comment 3: “Despite all of this, the results are presented clearly and in an organized manner, and some intervention proposals are derived. Here, again, it would be advisable to review the results obtained by other studies that have attempted to promote physical activity among adolescents, and perhaps adopt a more flexible perspective, considering other possible scenarios (Is it more necessary to try to encourage physical activity on weekends, or would it be more effective to promote it where it is already working, which is during the week?)”
Response 3: Thank you for pointing this out, we agree and added in the discussion section (page 16, lines 486-488) that the suggestions to effectively promote physical activity focus on emphasizing promoting weekend and designed for gender physical activities, based on our results and sample.